# Assessment of Urban Sustainability—The Case of Amman City in Jordan

**Firas M. Sharaf**

Department of Architecture, School of Engineering, The University of Jordan, Amman 11942, Jordan; f.sharaf@ju.edu.jo

**Abstract:** Accelerated urbanization causes an increasing number of city dwellers, insufficient and overburdened infrastructure and services, and negative environmental impacts and climate change impacts. Measuring the city's progress toward sustainability is essential to support decision-making and policy development. This study aims to establish an assessment and monitoring method of sustainable development goals at the city level, focusing on identifying indicators that are compatible with the city context to update and monitor progress toward sustainability. A review of the literature on sustainability assessment methods and tools is presented. A comprehensive framework for city sustainability assessment and a checklist of indicators. Amman city in Jordan is suggested. A Voluntary Local Review (VLR) report of Amman was presented to the United Nations in 2022. The report reviews Amman's progress toward achieving the SDGs; however, it lacks clear and a quantitative assessment of the city's sustainability, particularly SDG 11, which this paper seeks to address. The checklist survey questions were formulated according to the sub-indicators of the UN-Habitat SDG indicator metadata. The checklist was distributed to respondents from the Municipality of Amman and related organizations to the VLR. The respondents evaluated the sub-indicators of Goal 11 and gave performance level scores in three levels: low, average, and optimal. The sum of the indicator values is quantitatively presented in tables. The findings reveal that the indicator values of the city sustainability assessment framework, as applied in this paper, can be adjusted within the characteristics and constraints of the local context in a two-year observation period to provide updated data for decision-makers regarding the current status and future implementation of sustainability agendas.

**Keywords:** sustainable development goals; SDG 11; sustainability indicators; sustainability assessment; SDGs localization; Amman city

## 1. Introduction

Cities significantly impact the environment due to the activities that occur within cities and the circulation of products and individuals. By 2050, two thirds of the world's population will reside in urban areas, particularly in developing countries that are currently witnessing an increase in urbanization [1]. For example, urban areas in Jordan increased from 87.17% in 2011 to 91.63% in 2021. The urban challenge continues to increase as more people move to cities for a better life, economic opportunities, and refuge. The challenge is even greater for urban data collection and updating [2].

Cities have complex buildings and transportation systems and high consumptions of water, waste materials, energy, and electricity. The urban characteristics of city produce various environmental impacts. Hence, city sustainability is important [3]. A sustainable city is designed with consideration for social, economic, and environmental impacts and resilient habitats for existing populations without compromising the ability of future generations to experience the same [4]. A city needs to define how its residents can flourish while also ensuring that the aspirations of future generations are met through education, the development of city infrastructure, social and cultural development, and

preservation of the environmental [5]. Sustainability agenda developments in multiple sectors greatly influence the importance of implementing sustainable practices in cities and local neighborhoods. Efforts to address the sustainability of city sectors help build communities that are immersed in sustainability practices [6].

Long-term sustainability considerations are essential for more than environmental protection. This approach acknowledges that investing in human, natural, and intellectual capital is frequently neglected in favor of an emphasis on economic or financial value [7,8]. Although different strategies for conducting sustainable practices have been proposed by many cities, measuring the progress of practices toward sustainability is also essential to support decision-making and policy development and measure environmental impacts and other contexts [9].

A sustainability assessment provides short- and long-term insights into a city and its natural and social systems to help make decisions about actions that should or should not be taken to manage a sustainable community [10]. Sustainability valuation indicators and composite indicators are increasingly recognized as valuable tools for policy-making and mass communication because they convey information about the performance of cities and regions in areas such as the environment, economy, society, and technological innovation [11,12].

An international strategy provides national and local guidelines for the development and expansion of cities which promotes sustainable development as a central issue in the pursuit of the SDGs [13]. The United Nations published three sets of sustainable development indicators in 1996, 2001, and 2007 which highlight the importance of developing indicators at city and regional levels that align with national conditions and priorities [14]. The integration of indicator initiatives with national development policies and their conversion into permanent programs of action ranked high among the recommendations of the United Nations [15].

## 2. Indicators of Sustainable Development

Sustainability for cities require carefully tailored methods to assess urban sustainability performances according to their local or national context. The merit of the assessment methods is evaluated in practice in cities [16,17]. Several sets of indicators for urban sustainability are identified in the literature; for example, energy-based indicators of ecosystem services in Simao city in China [18]; a green spaces assessment approach to health, safety, and the environment [10]; construction and city-related sustainability indicators in Europe [19,20]; infrastructure sustainability indicators in South African [21]; and smart, sustainable city indicators [22–24]. No single set of indicators applies to all cities or communities equally [18,25]. However, it was realized that using standard indicators would facilitate tracking and comparing urban sustainability processes so that it would not continue to be an ethereal idea [26].

Establishing comparable indicators may enable cities to share and disseminate practical tools and create a shared network that focuses on the city scale as the level where the application of urban sustainability indicators can be best appreciated and compared [19,27].

Indicators have a role in measuring urban sustainability performance; hence, there is a need for measurable indicators. Several approaches to assess urban sustainability-based indicators have been developed [19]; for example, using methodological foundations of different evaluation methods to propose classifications divided into different groups, such as system engineering, critical evaluation, and biophysics [28]. Other approaches document the extent to which cities are becoming or not becoming sustainable through indicators and reveal practical challenges encountered in the process [29–33]. The indicator selection process should not focus on gathering information for all indicators but instead on selectively assessing those most important and most likely to generate accurate and reliable data on the current state of practice [32,33]. Indicator characteristics should be clear, simple, scientifically sound, verifiable, and repeatable [34]. Other researchers proposed that urban sustainability indicators should provide at least three tools: explanatory tools to translate

the concepts of sustainable development into practical terms, pilot tools to assist in making policy choices that promote sustainable development, and performance assessment tools to decide how effective the efforts have been [35]. The UN-SIAP defines sustainability indicators as SMART, which means the indicators should be specific, measurable, achievable, relevant, and time-related [36].

United Nations organizations have provided a comprehensive and common framework for monitoring the achievements of sustainable urban development worldwide. The UN-Habitat used a global tool to measure sustainable development in the City Prosperity Initiative (CPI) which applied measurable indicators and policy discourse. The CPI proposed a six-dimensional composite index to identify goals and objectives that can support evidence-based policy formulation, including defining city visions and long-term plans that are ambitious and measurable [37]. The systematic framework of the National Urban Policy (NUP) has been utilized by UN organizations to monitor and evaluate sustainable development progress at the international level with commonly defined methodologies and processes. The objective is to provide policymakers, practitioners, and academia with evidence and country-level experiences [38]. The United Nations report "The Global State of National Urban Policy—2021" lays the groundwork for understanding why and how to develop, implement, and monitor city policies globally. One may use the links between the National Urban Policy (NUP) and the global SDGs to assist cities in localizing the SDGs and designing frameworks for sustainability assessment and monitoring [39]. Therefore, the question arises of what methods are best to implement and assess sustainability issues at a local city level [40].

This paper aims to investigate methods for assessing and monitoring the sustainability of cities and communities based on SDG indicators. Existing methods and tools for city assessments are first briefly reviewed.

### 3. Methods and Tools for Assessing the Sustainability of Cities

Current information, data, and analyses of urban contexts and trends are crucial to understand the state of cities and reinforce their sustainability. Several methods and tools for assessing the sustainability of cities and neighborhoods have been developed [41,42], such as technical reports issued by the International Organization for Standardization (ISO) [43,44]. However, current sustainability assessment, monitoring, and management circumstances require further investigation. Interagency and expert groups have proposed many indicators to support the efforts of stakeholders to monitor the global progress of SDGs [45]. Nevertheless, the proposed indicators aim to monitor the progress of the SDGs at the international level. They, therefore, include some indicators that have no practical use for monitoring progress at the local level. Indicators should be adapted, or alternative indicators should be anticipated to monitor the efforts of local administrations and the achievements of the SDGs at the city level. Although comprehensive sets of indicators have been developed for assessing cities [46], a valuation method for tracking SDG progress at the city level is still lacking, which would support local efforts and initiatives in cities and urban areas [47].

This study aims to properly understand the assessment and monitoring of sustainable development goals at the city level, focusing on the localization of SDG indicators and the identification of indicators that are compatible with the sustainable development goals [48,49].

### 4. Sustainability Assessment Context for the Case Study of the City of Amman

Amman is a capital city located in northwest Jordan with an estimated population of 4.5 million people (13.5% of the total population of Jordan) and an average density of 25,000 people/km$^2$. Amman's urban area is 630 km$^2$, with an annual increase rate of 2%, which has increased pressure on municipal public services. In 2019, the City of Amman launched the Green City Action Plan (GCAP) to define the city's priorities for addressing climate change and environmental challenges, including solid waste management, water

and wastewater management, urban transport, and building energy efficiency, in order to reduce local pollution, improve energy and resource efficiency, and promote adaptation to climate change [50].

The Greater Amman Municipality (GAM), collaborating with the United Nations agencies of UNESCWA and the UN-Habitat office in Jordan, prepared a sustainability assessment report for Amman's Voluntary Local Review (VLR). Amman's VLR focuses on the city's progress toward SDGs 3, 7, 9, 11, 13, and 17. The goals and targets reviewed were selected based on their relevance to the mandate, the municipality's strategic objectives, and access to and availability of data. Special attention is made to mainstreaming gender considerations and discussing refugee challenges. Amman's progress in achieving SDG11 is presented in Chapter 2 of VLR which addresses two indicators, including housing and transportation, and two sub-indicators of the city's environmental impact, including waste management and air quality. The Amman VLR needs a more precise and accurate quantitative assessment of the indicators [51].

This paper addresses this deficiency and proposes a methodology to assess each of the eleven SDG indicators and provides a more comprehensive sustainability assessment.

## 5. Method

*Developing a City Sustainability Assessment Method*

A Framework for Assessing Sustainability in Amman City

Assessment indicators were proposed to quantify the sustainability assessment of Amman city. To this end, candidate indicators were determined following a review of the materials issued by the institutions that produced the VLR of Amman city. These institutions include the Municipality of Amman, UN-Habitat, and UNESCWA. SDG11 is most relevant to urban development that focuses on cities as its focus is to "make cities and human settlements inclusive, safe, resilient, and sustainable, containing nine categories" [52,53]. A checklist survey was created according to the nine indicators of SDG 11 to comprehensively assess the city of Amman's sustainability. The survey questions in Appendix A were formulated according to each sub-indicator item and subtracted from the SDG indicator metadata 2021 and the World Cities Report 2022 [54–56]. The evaluation of each item consisted of three performance levels: low, average, and optimal. The checklist was distributed to ten key informants from the Municipality of Amman, the UN-Habitat Jordan office, UNESCWA, and Amman Urban Observatory, who were all familiar with the VLR 2022 of Amman. The evaluation of each indicator was dependent on the value of the sub-indicators. The respondents were asked to evaluate the sub-indicators of the nine indicators of Goal 11 and give scores to match the situation in the city and evaluate the level of performance in the three levels or leave the questions blank in order to reduce error. The indicator values were calculated following data collection, and the initial values were organized into tables and aggregated for each indicator. The sum of the responses for each sub-indicator was placed in the designated level under L, A, or O (Appendix A). For example, if there were three sub-indicators, six replies were in the low-level box of the first sub-indicator, four in the low-level box of the second sub-indicator, and two in the third, the average would be 6 + 4 + 2 = 12 ÷ 3 = 4. A similar procedure to calculate averages for levels A and O was conducted. The sum of all values of the indicators is presented in Table 1.

**Table 1.** The sum of the values of the indicators to assess the sustainability of Amman city.

|  | L | A | O |
|---|---|---|---|
| 1. Accessibility of suitable, secure, and affordable housing, essential services, and shanty areas improvement | 4 | 4 | 2 |
| 2. Public transportation | 7.5 | 7 | 4.5 |

**Table 1.** *Cont.*

| | L | A | O |
|---|---|---|---|
| 3. Impact on the environment, water sustainability, air quality, and waste management | 8.6 | 1.8 | 0.44 |
| 4. Preserving and protecting the natural and cultural heritage of Amman | 6.6 | 2.7 | 0.7 |
| 5. Access and delivery of secure green areas and public spaces | 6.7 | 3.4 | 1 |
| 6. Incorporated resource proficiency and adaptation to climate transformation and natural catastrophes | 5.3 | 3.7 | 1 |
| 7. Create durable and flexible buildings using adaptable local materials and sustainable locations | 6.3 | 3 | 3.3 |
| 8. Resilient urban development, integrative and sustained neighborhood design, and administration capability | 6.5 | 3 | 0.7 |
| 9. Reinforce national and local development plans to strengthen socioeconomic and environmental linkages between cities, suburbs, and county areas | 5.2 | 4.5 | 0.4 |
| The sum of level scores | 56 | 33 | 15 |

## 6. Result and Discussion

Table 1 presents the results of the sustainability assessment survey checklist which is provided in Appendix A.

1. The assessment result of the first indicator—access to decent, safe, and inexpensive housing, essential utilities, and the improvement of shantytowns is average-low; however, the proportion of the population living in slum households is less than the country's proportion of 18% [57]. The assessment of the eligible population ratio included in the national social security is low-average. The assessment level of encounters with a qualified professional in a medical facility or the community per capita per year is average-low. The level of residents using safely managed water services in urban areas is relatively average as most households are connected to Amman's main municipal water supply. In contrast, the level of residents using securely managed sanitation facilities is low as some houses are not connected to the main sanitation facility. The ratio of residents using modern cooking solutions is average as cooking solutions are available in markets at affordable costs.

2. In general, the evaluation of the public transportation performance level in Amman is low, especially for the sub-indicators that assess the regularity of journey time, fares, and the collection system. The public transport network does not cover all urban areas of the city, and its employees show poor performance. There is a low sustainability level of the current transportation fleet in terms of vehicles powered by green energy.

3. The environment impact assessment level is low, including the sub-indictors of water sustainability, air sustainability, and waste management. However, the assessment of the sub-indicator of water pollution reduction is average because special attention is made to the water quality of Amman. Houses have access to a municipal water supply once per week, and outages can last for several weeks, especially in the long, dry summer season. Additional water tanks are needed for houses to maintain water supplies during water cut-offs, and houses must purchase water when it runs out. In addition, water leakage from the principal water network is high, and households suffer from water scarcity [58]. Domestic garbage is disposed of in landfill sites, contaminating the soil and the underground water, and the city lacks efficient garbage and recycling systems. Air pollution in Amman results from human activities, especially from the transportation, industry, and energy sectors where fossil fuels are burned, in addition to dust storms and airborne pollutants. Although the quality of air in Amman is, on average, within the limits allowed by national standards, the limits in the Jordanian national standards for air quality are higher than the WHO guideline limits [57].

4. The assessment of the conservation and protection of natural and cultural heritage levels needs to be higher. Amman has solid historical and contemporary heritages and natural assets, yet it needs to adequately employ the potential of its heritage to advance community development and economic progress. This heritage is evident in the low assessment of indicators, such as the contribution of natural and cultural heritage to eliminate poverty, sustainable food production and consumption, the agricultural landscape, healthy lives, sustainable food production and consumption, the waterscape, the rural and agricultural landscape, the sustainable management of water resources, and energy-efficient forms of growth.

5. Public parks in Amman city have average accessibility, safety, and inclusivity levels. The city provides 2.5 square meters of park area per person which is 28% of the minimum 9 $m^2$ per capita [59]. The availability of public parks within a standard radius is low as only 30% of Amman city areas are within service distance of public parks. This finding is illustrated in Table 2 and Figure 1. Appendix A, Table A1 (5) shows low scores in car parking inclusivity, external stairs and ramps, general training, and emergency preparations within public parks. This indicates neglect to follow design standards for implementing public parks and the provision of required metropolitan or regional parks [60,61].

**Table 2.** Classification of urban parks [62].

| Classification | Size | Service Radius |
|---|---|---|
| Mini park | Less than 1.5 acres (15,000 $m^2$) | Less than 402 m (sub-neighborhood) |
| Neighborhood park | Optimum size 50,000 $m^2$–150,000 $m^2$ | A 0.25–0.5-mile distance uninterrupted by nonresidential roads (402–804 m) |
| Community park | 16–99 acres 160,000 $m^2$–990,000 $m^2$ | 1–2 miles and serves two or more neighborhoods (804–4828 m) |
| Metropolitan park | 100–499 acres 1,000,000–4,990,000 $m^2$ | Within 30 min driving time (20–30 km 12–18 miles) and serves the entire city |
| Regional park | More than 500 acres (5,000,000 $m^2$) | Within one hour's driving time |

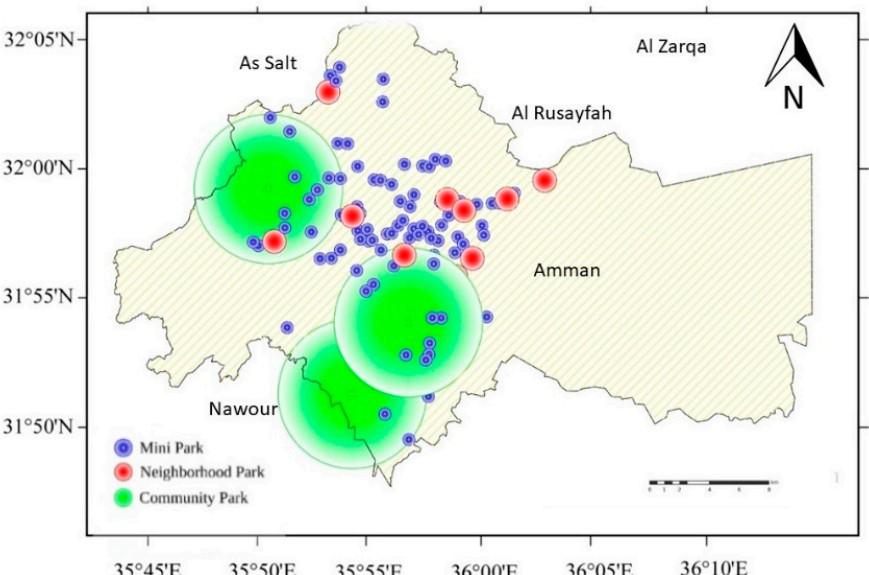

**Figure 1.** Amman's public park service radius [63].

6. The assessment results of the sub-indicators of inclusion, resource efficiency, and resilience toward climate change and disasters indicate that readiness to combat climate change and disasters in Amman city is mostly of low performance. There are high levels of unemployment, stalled businesses, economic investment challenges, and low income levels for many families.

However, during the COVID-19 pandemic, resilience to disaster levels has increased through better inter-department collaboration regarding disaster preparedness and risk reduction measures, updated information and evaluation, and upgraded medical facilities. Recently, the collapse of a building in Al Wabda in Amman initiated the implementation and imposition of real, risk-acquiescent building laws. There are investment and start-up business challenges, low family income levels, and limited youth and women empowerment opportunities. Although there are various energy sources in Amman, such as solar cells and electric vehicles, their proportion still needs to be improved. The application of the national green building code is limited—only 0.5% of Amman's buildings apply restricted items, such as thermal insulation and solar water heaters. Amman city encounters an extreme shortage of domestic water supply and needs more sustainable solutions, such as rainwater harvesting and gray water systems. While solid waste management, including collection and transportation, is provided in the city, treatment in landfill sites causes land, air, and underground water pollution.

7. Building sustainable and resilient structures through utilizing local materials scores low in material reuse, construction waste management, and the collection of recyclable building materials as they have low-level applications. Low scores are recorded in energy efficiency items, such as building orientation, roofs, walls, shading devices, ventilation access, and gray water systems. Sustainable building methods and materials need to be more effectively applied in architectural design and construction to be compatible with the expected sustainability levels. Although there are incentive regulations and codes for green buildings and sites in Jordan, the high cost of materials reduces the prevalence of green buildings and sustainable sites in Amman city.

8. Sustainable urbanization and neighborhood planning includes four low scores of nine in environmental protection and eight low scores of ten in economic development and the social justice and equity items. Poor environmental, economic, and social performances are caused by a lack of effective zoning regulations, urban service boundaries, neighborhood planning, and living wage ordinances. There are few affordable provisions in housing, city public transit, and farming and little clustered or targeted economic development.

9. Results regarding city links with peripheries, suburbs, and province areas show average to low scores in reinforcing national and local development plans. Items such as investments in urban–rural development and combined planning through the municipal-rural spectrum have average scores. Governance, legislation, capacity development, and the management of land and natural resources have low scores and need serious attention to accomplish better social, economic, and environmental linkages.

Finally, the total scores of the overall indicators of Amman city include 56 in the low performance level, while 33 is the average, yet 15 is optimal. This score is relatively low–moderate. However, more attention should be made to the city's environmental impact and water sustainability.

## 7. Conclusions

The city sustainability assessment method was explored in this study to allow for more comprehensive and numerous assessment indicators. Its purpose was for users, such as local government officers, citizens, and other stakeholders, to understand the actual conditions of their cities and make progress toward achieving global SDGs.

In total, 154 sub-indicators were evaluated from the perspective of the level of performance and compared to assess their performance level, feasibility, and accuracy of the evaluation results. Indicator assessments were conducted using a key informant survey.

However, a larger survey sample of users and residents would provide more accurate results, but time and cost constraints limited doing so in this study.

An ideal development path toward sustainable cities is alleviating indicators of values lower than average by scaling up their quality and redirecting resources to that goal while also maintaining the quality of other indicators. In addition, there should be a focus on improving quality without increasing the environmental burden. It is recommended that the city establish a monitoring period to assess improvements. For cities to understand their current sustainable conditions, a comparison can be made to other cities in the state. The assessment method proposed in this study is expected to assist future sustainability assessments in moving toward an increasingly sustainable urban future for all residents in Amman and could be applied to other cities (with some modifications according to the local context).

Jordan provides baseline data from its regional and country levels for the global assessments undertaken every two years to inform international UN agencies of the State of National Urban Policy (NUP). This data has been updated yearly since 2018.

The monitoring and reporting of sustainability indicators should be conducted at annual intervals. Comprehensive reporting must be undertaken once every two years and made available online on urban policy platforms.

**Funding:** No funding was provided to conduct this study.

**Informed Consent Statement:** Not applicable.

**Data Availability Statement:** Data is unavailable due to privacy or ethical restrictions.

**Acknowledgments:** Thank you to everyone who contributed to the preparation of this paper, particularly Suhaib Abuhazeem, Maria Bani Hani, Aseel Obaid, Shahrazad Ghreir, Eman Naji, Sara Jawabreh, Mohammad Khreisat, Ahmad Salem, Amal Al-Hourani, and Leen Shadid.

**Conflicts of Interest:** The author declares no conflict of interest.

## Appendix A. Sustainability Assessment Checklist for the City of Amman

**Table A1.** Scores: mark $\sqrt{}$ L □ if the item availability is low, A □ if it is average, and O □ if it is Optimal Respondents were instructed to leave items blank if they were not available. n = 10. The scores of each item are placed next to the box (e.g., □5 means 5 respondents marked this level).

| 1. Convenient accessibility to suitable, secure, and inexpensive housing, essential services, and slum improvement | low L | average A | optimal O |
|---|---|---|---|
| Eligible population ratio included in the national social security | $\sqrt{}$□5 | $\sqrt{}$□4 | $\sqrt{}$□1 |
| Encounters with a qualified professional at a medical facility or in the community per capita per year | □4 | □4 | □2 |
| Ratio of residents using securely managed water facilities | □4 | □5 | □1 |
| Ratio of residents using securely managed sanitation services | □5 | □3 | □2 |
| Ratio of residents using contemporary cookery facilities | □2 | □5 | □3 |
| Average of the scores | 20 ÷ 5 = 4 | 4 | 2 |
| **2. Public transport** | | | |
| Facilities at the station (protection from weather, lighting, etc.) | □3 | □4 | □3 |
| Cleanliness of vehicles | □5 | □4 | □1 |
| AC availability | □10 | □ | □ |
| Privacy | □9 | □1 | □ |
| Seat comfort | □8 | □2 | □ |
| Noise | □8 | □2 | □ |

| | | | |
|---|---|---|---|
| Crowding | ☐9 | ☐1 | ☐ |
| Availability of service | ☐6 | ☐3 | ☐1 |
| Availability of wheelchair space | ☐10 | ☐ | ☐ |
| Ease of entering/exiting vehicles | ☐9 | ☐1 | ☐ |
| Ease of reimbursement | ☐5 | ☐4 | ☐1 |
| Service of transportation network (quantity of routes) | ☐4 | ☐5 | ☐1 |
| Traveling expenses | ☐7 | ☐2 | ☐1 |
| Staff behavior | ☐7 | ☐2 | ☐1 |
| Journey time | ☐7 | ☐2 | ☐1 |
| Waiting time | ☐8 | ☐2 | ☐ |
| Safety in the vehicle | ☐5 | ☐3 | ☐2 |
| Personal security for females | ☐8 | ☐2 | ☐ |
| Average scores | 7.5 | 2.35 | 0.88 |
| **3. Impacton the environment,water sustainability, airquality, andwaste management** | | | |
| Balancing water supply and demand | ☐8 | ☐2 | ☐ |
| Re-evaluating water value | ☐7 | ☐2 | ☐1 |
| Application of floating discs | ☐9 | ☐1 | ☐ |
| Rain harvesting | ☐9 | ☐1 | ☐ |
| Public education campaigns | ☐9 | ☐1 | ☐ |
| Reuse of water | ☐9 | ☐1 | ☐ |
| Developing infrastructure | ☐7 | ☐2 | ☐1 |
| Reduction of pollution in the water | ☐1 | ☐5 | ☐4 |
| Get rid of some harmful gases in the water in a natural way | ☐8 | ☐1 | ☐ |
| **Air quality** | | | |
| Strong consequences for emissions | ☐8 | ☐2 | ☐ |
| Controlled air quality | ☐7 | ☐3 | ☐ |
| Planting concertation | ☐5 | ☐3 | ☐1 |
| Reusing bad gases | ☐10 | ☐ | ☐ |
| **Waste management** | | | |
| Waste separation | ☐8 | ☐2 | ☐ |
| Recycle waste | ☐9 | ☐1 | ☐ |
| Public education campaigns | ☐9 | ☐1 | ☐ |
| Consequences for factories | ☐8 | ☐2 | ☐ |
| Average sum of scores | 8.6 | 1.8 | 0.4 |
| **4. Preserving and protecting thenatur al and cultural heritage of Ammancity** | | | |
| Harness the influence of heritage to eliminate prevalent poverty for all | ☐7 | ☐2 | ☐1 |
| Utilize tradition: sustainable food production and consumption, the waterscape, rural areas, the agricultural landscape, and intangible and biotic heritage | ☐6 | ☐3 | ☐1 |
| Harness the potential of heritage, ensuring healthy lives, and promoting the wellbeing of residents | ☐7 | ☐2 | ☐1 |

| | | | |
|---|---|---|---|
| Harness the influence of heritage to support access to quality education for all residents of the city | ☐8 | ☐2 | ☐ |
| Employ the potential of heritage to advance equality and empowerment and eradicate bias and violence between the sexes, recognizing that tradition is evolving continuously | ☐6 | ☐3 | ☐1 |
| Facilitate the potential of heritage by providing viable plans for the sustainable management of water resources that ensure everyone has access to clean water and sanitary facilities | ☐6 | ☐3 | ☐1 |
| Facilitate the ability of heritage to contribute to energy-efficient forms of growth | ☐7 | ☐3 | ☐ |
| Use heritage as a resource for effective, comprehensive, and sustainable micro-and macro-economic growth | ☐7 | ☐3 | ☐ |
| Utilize heritage for inclusive and continued manufacturing and infrastructure through inspiration and novelty | ☐7 | ☐3 | ☐ |
| Harness heritage tasks by decreasing inequality and promoting inclusivity and cultural variety | ☐6 | ☐3 | ☐1 |
| Incorporate heritage as a catalyst and inspiration for durable production and consumption | ☐6 | ☐3 | ☐1 |
| Harness heritage to enhance the ability of societies to change, adapt, and build resilience in the face of climate change | ☐7 | ☐3 | ☐ |
| Harness the power of heritage to combine value-based, landscape-based, and human right-based methods to protect, restore, and sustain ecosystems | ☐6 | ☐3 | ☐1 |
| Enhance the role of heritage in building just, comprehensive, and peaceful communities | ☐7 | ☐2 | ☐1 |
| Promote sustainability-oriented heritage development policies and practices by harnessing the potential of strategic partnerships in heritage operations | ☐6 | ☐3 | ☐1 |
| Average sum of scores | 6.6 | 2.7 | 0.66 |
| **5.Access and delivery of secure green areas and public spaces** | | | |
| **Parks safety** | | | |
| Park furniture safety | ☐5 | ☐4 | ☐1 |
| Entries and paths safety | ☐5 | ☐4 | ☐1 |
| Spaces allocation | ☐7 | ☐2 | ☐1 |
| Materials appropriateness with activities | ☐5 | ☐4 | ☐1 |
| Adequate lighting | ☐4 | ☐5 | ☐1 |
| Proper use of signs | ☐4 | ☐5 | ☐1 |
| Public training and emergency preparations | ☐9 | ☐1 | ☐ |
| Security | ☐3 | ☐5 | ☐2 |
| Playground immunization | ☐3 | ☐5 | ☐2 |
| **Inclusive design of parks** | | | |
| Inclusivity of car parking | ☐7 | ☐2 | ☐1 |
| Inclusivity of access routes and wayfinding | ☐4 | ☐5 | ☐1 |
| The choice of surface materials | ☐7 | ☐2 | ☐1 |
| Inclusivity of external staircases and ramps | ☐5 | ☐4 | ☐1 |
| Inclusivity of park furniture | ☐7 | ☐2 | ☐1 |
| Inclusivity of sanitary facilities | ☐7 | ☐2 | ☐1 |

| **Accessibility and standard availability of parks** | | | |
|---|---|---|---|
| Availability of public transportation | ☐4 | ☐5 | ☐1 |
| Availability of suitable parking spaces | ☐7 | ☐2 | ☐1 |
| Entry and circulation routes | ☐4 | ☐5 | ☐1 |
| The area within the service radius of public parks | ☐7 | ☐2 | ☐1 |
| Park area per capita | ☐9 | ☐1 | ☐ |
| Average sum of scores | 5.65 | 3.35 | 1 |
| **6. Incorporated resource proficiency and adaptationto climatechangeand natural disasters** | | | |
| **Policies and plans toward inclusion** | | | |
| Leverage existing human capital to create employment | ☐5 | ☐3 | ☐2 |
| Support entrepreneurs and start-up businesses | ☐5 | ☐4 | ☐1 |
| Empower women | ☐4 | ☐5 | ☐1 |
| Integrate and engage young people equally and culturally | ☐5 | ☐3 | ☐2 |
| **Resource efficiency** | | | |
| Improve energy security by diversifying energy sources (e.g., solar cells, electric vehicles, etc.) | ☐5 | ☐3 | ☐2 |
| Apply green building codes and guidelines | ☐5 | ☐3 | ☐2 |
| Manage water resources efficiently | ☐4 | ☐5 | ☐1 |
| Improve the waste management system | ☐5 | ☐5 | ☐ |
| **Mitigation and adaptation to climate change** | | | |
| Strengthen and enhance natural disaster preparedness and resilience to climate-related risks | ☐6 | ☐3 | ☐1 |
| Create a climate change action plan and incorporate climate change policies into national strategies. | ☐6 | ☐3 | ☐1 |
| Boost human and institutional capacity for mitigation, adaptation, impact reduction, and early warnings of climate change | ☐6 | ☐4 | ☐ |
| **Resilience to disasters** | | | |
| Ensure that departments provide appropriate organization and collaboration to improve disaster preparedness and risk reduction measures | ☐7 | ☐3 | ☐ |
| Keep information on risks and weaknesses updated | ☐3 | ☐5 | ☐2 |
| Evaluate the security of all educational and medical facilities and make any necessary upgrades | ☐6 | ☐3 | ☐1 |
| Implement and impose real, risk-acquiescent building laws | ☐6 | ☐3 | ☐1 |
| Implement and impose real, risk-acquiescent land use development guidelines | ☐6 | ☐4 | ☐ |
| Average sum of scores | 5.3 | 3.7 | 1 |
| **7. Create durable and flexible buildings using adaptable local materials and sustainable locations** | | | |
| Protect the surrounding environment | ☐8 | ☐2 | ☐ |
| General safety | ☐7 | ☐3 | ☐ |
| Green areas | ☐7 | ☐3 | ☐ |
| Roof area | ☐8 | ☐2 | ☐ |
| Transportation | ☐7 | ☐3 | ☐ |
| Open areas | ☐7 | ☐3 | ☐ |

| Water efficiency | | | |
|---|---|---|---|
| Water saving devices | ☐7 | ☐3 | ☐ |
| Pipes insulation | ☐8 | ☐2 | ☐ |
| Rainwater harvesting | ☐8 | ☐2 | ☐ |
| Gray water systems | ☐9 | ☐1 | ☐ |
| **Energy efficiency** | | | |
| Orientation of the building | ☐10 | ☐ | ☐ |
| Roofs and walls of the building envelope | ☐8 | ☐2 | ☐ |
| Thermal insulation of the building envelope | ☐6 | ☐4 | ☐ |
| Day illumination | ☐6 | ☐4 | ☐ |
| Shading devices | ☐3 | ☐5 | ☐2 |
| Air ventilation | ☐3 | ☐4 | ☐3 |
| Lighting | ☐3 | ☐4 | ☐3 |
| **Healthy indoor environment** | | | |
| Thermal comfort | ☐3 | ☐5 | ☐2 |
| Lighting efficiency | ☐3 | ☐5 | ☐2 |
| Access to ventilation | ☐3 | ☐5 | ☐2 |
| **Materials and resources** | | | |
| Green materials | ☐3 | ☐5 | ☐2 |
| Materials reuse | ☐8 | ☐2 | ☐ |
| Construction waste management | ☐8 | ☐2 | ☐ |
| Collection of recyclable building materials | ☐8 | ☐2 | ☐ |
| Average sum of scores | 6.3 | 3 | 0.66 |
| **8.Resilienturban development, integrative and sustainable neighborhood design, and administration capability** | | | |
| **Environmental protection** | | | |
| Alternate power provided to users | ☐3 | ☐4 | ☐3 |
| Energy preservation endeavor (plus green construction requisites) | ☐6 | ☐4 | ☐ |
| Eco-friendly site planning laws | ☐7 | ☐3 | ☐ |
| Green construction plans | ☐8 | ☐2 | ☐ |
| Pavement side reusing schemes | ☐6 | ☐4 | ☐ |
| Ecofriendly learning systems for residents | ☐8 | ☐2 | ☐ |
| Safeguarding the quality of water | ☐3 | ☐5 | ☐2 |
| The running of inter-city communal transportation (buses and trains) | ☐3 | ☐5 | ☐2 |
| The management of transport requirements | ☐3 | ☐5 | ☐2 |
| **Economic development** | | | |
| 10. Subdivision for farmland preservation | ☐6 | ☐4 | ☐ |
| Rehabilitation of contaminated sites | ☐6 | ☐4 | ☐ |
| Focused or clustered economic progress | ☐7 | ☐3 | ☐ |
| Planning green industrial zoning | ☐8 | ☐2 | ☐ |
| Transmission or acquisition of progress entitlements | ☐8 | ☐2 | ☐ |
| Tax breaks for ecofriendly construction | ☐9 | ☐1 | ☐ |

| | | | |
|---|---|---|---|
| Municipal development edge or urban amenity limits | ☐5 | ☐3 | ☐2 |
| Business preservation program | ☐8 | ☐2 | ☐ |
| Infill management | ☐2 | ☐7 | ☐1 |
| Empowerment/enterprise zones | ☐5 | ☐3 | ☐2 |
| **Social justice and equity** | | | |
| Provide low-cost housing | ☐2 | ☐6 | ☐2 |
| Childcare for service segment and limited-income workers | ☐9 | ☐1 | ☐ |
| Interference programs to prevent homelessness | ☐9 | ☐1 | ☐ |
| A balance between jobs and housing | ☐8 | ☐2 | ☐ |
| An adequate living income law | ☐9 | ☐1 | ☐ |
| Accessibility to public transportation through revenue support | ☐9 | ☐1 | ☐ |
| Community design | ☐9 | ☐1 | ☐ |
| Programs for durable food production and supply systems | ☐8 | ☐2 | ☐ |
| Support for local investments and women and minority-owned businesses | ☐8 | ☐2 | ☐ |
| The creation of more opportunities for youth and a gang prevention scheme | ☐7 | ☐3 | ☐ |
| Average sum of scores | 6.5 | 3 | 0.66 |
| **9. Reinforce national and local development plans to strengthen socioeconomic and environmental linkages between cities, suburbs, and county areas** | | | |
| Capacity building, lawmaking, and administration | ☐3 | ☐6 | ☐1 |
| Integrated planning across the municipal–rural spectrum | ☐7 | ☐3 | ☐ |
| Investing and financing for comprehensive municipal–rural advancement | ☐8 | ☐2 | ☐ |
| Community empowerment | ☐3 | ☐6 | ☐1 |
| Managing expertise and data for the dynamical geographical mobility of people, goods, facilities, resources, and information | ☐7 | ☐3 | ☐ |
| Developing the local economy and creating job opportunities | ☐4 | ☐6 | ☐ |
| Cogent methods for providing social services | ☐7 | ☐3 | ☐ |
| Systems for infrastructure, technologies, and communications | ☐3 | ☐6 | ☐1 |
| Cohesive methods for food and agricultural production and general healthiness | ☐3 | ☐6 | ☐1 |
| Ecological effects and the management of natural capital and terrestrial | ☐6 | ☐4 | ☐ |
| Communication between urban and rural areas to confront disputes and calamity | ☐6 | ☐4 | ☐ |
| Average sum of scores | 5.2 | 4.45 | 0.36 |

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
