# Peer review of "Assessment of Urban Sustainability—The Case of Amman City in Jordan"

_sustainability, doi:10.3390/su15075875_

Round 1

Reviewer 1 Report

The topic of the paper is current. However, significant shortcomings in its content were noticed.

- lines 79 and 84: It says "Five types ..." and it says ... 6 types, because in line 84 there are two types together, i.e. "Human capital" and "Manufactured capital";

- there is a lack of a broader review of the literature in terms of other similar case studies and the methods that are used during such assessments;

- in section 3. Methods there is no information on what the questions in the questionnaire looked like?, how many respondents answered these questions (what was the size of the research sample)?, what technique was used to conduct the research? how were respondents selected? what is the credibility of the respondents, i.e. how was their knowledge and competence assessed during the research? how was the score (low, medium, high) determined from the responses received in the questionnaires?

- there is no justification that the presented results bring new knowledge to the current state of science.

Author Response

lines 79 and 84: It says "Five types ..." and it says ... 6 types, because in line 84 there are two types together, i.e. "Human capital" and "Manufactured capital";

- More literature has been add to the revised copy.-there is a lack of a broader review of the literature in terms of other similar case studies and the methods that are used during such assessments;

- more clarification is in the revised copy sent---in section 3. Methods there is no information on what the questions in the questionnaire looked like?, how many respondents answered these questions (what was the size of the research sample)?, what technique was used to conduct the research? how were respondents selected? what is the credibility of the respondents, i.e. how was their knowledge and competence assessed during the research? how was the score (low, medium, high) determined from the responses received in the questionnaires?

- a justification has been make in the revised copy sent --there is no justification that the presented results bring new knowledge to the current state of science.

Reviewer 2 Report

Dear Authors,      

The paper entitled " Assessing Urban Sustainability: The Case of Amman city in Jordan" aims to identify the characteristics and levels of urban sustainability, and addresses the issue from an urban development perspective to achieve cities that are "inclusive, safe, resilient and sustainable." 

The research topic of the article is interesting because it addresses the important issue of urban sustainability. The article creates a series of social and economic indicators to assess the level of sustainability. However, in order for the article to be published in Sustainability it needs a number of corrections. 

After reading the article, I have the following comments and suggestions for improving the article:    

Abstract    

I propose to improve it by making it more readable.  I suggest improving the abstract according to the Journal "Sustainability" guidelines. There is no information about the methods used and the results of the study are not presented.    

In the Introduction    

This chapter needs to introduce the theoretical background based on the literature. In my opinion, it should be expanded to include the following news: why was this study undertaken?  What studies have been conducted so far, where? What conclusions have been drawn from these studies? Is this article a continuation of those conclusions, or is it based on your own observations?   

In Material and Methods    

A detailed description of the methodological process is lacking. There is no information about the study area. There is no information on what criteria guided the selection of the scale . 

A diagram of the research procedure is missing.   

Result.    

In this chapter there is a very long table without commentary. The commentary on the table is in the Discussion section. I suggest organizing the content of the article into the appropriate chapters. 

Technical errors to be removed:  

There is too much numbering in the text, I suggest using different characters. 

Please correct the literature according to the rules of the journal.  

Regards, 

Reviewer

Author Response

major revisions are made and  your notes are addressed

Reviewer 3 Report

The paper deals with the process of assessing the sustainability of cities – this is an interesting and trending subject – the fact the city of Amman is used as a use case makes it even more special. However, there are several major shortcomings that need to be resolved before publishing this work. Please find below my detailed comments:

Abstract:

I do not see the added value of the first lines (9-15) to appear on abstract - since this is general background information that better fits in Introduction. I suggest to discard this info and instead strengthen the methodology and results related info in the abstract (e.g., criteria extracted, info on checklist, score-results).

The text requires some good polishing on the use of English – there are several syntactical and grammatical errors here and there e.g., line 16, 23 – the same applies for the rest of the document.

1. Introduction:

Introduction needs to be significantly enhanced. Author should clearly define what is meant by the term sustainability and especially sustainable city (some hints are available in Section 2). I would expect to see a much more detailed reference to similar works available in literature, including already available tools to assess sustainability and quick reference to results (which cities evaluated etc.). The author is mentioning that ‘Although many cities have proposed different strategies to implement sustainable practice, yet, the degree of these practices need to be deliberate’. This needs to be justified. Why is there a need to develop a method to assess a city’s sustainability? How about current methods – what are their pros and cons?

Please also note that the vast majority of references are before 2020, thus I highly recommend the author to also check more updated works on this subject.

2. Indicators of city sustainability

I do not understand the scope of Section 2 (why a separate section is there). It makes sense for example if the author plans to select, extract etc. his/her own KPIs but the flow of information is not very clear – the story telling needs to be enhanced (e.g., why the five types of value are mentioned in line 79? Why the Green City Index is specifically described? – and not any other of the various similar Indexes available?)

Since the evaluation is based on the nine indicators as imposed by the SDG Goal 11, it is very important to justify and better support the reasoning behind this choice.

3. Method

This section needs significant improvement as well. The author mentions three parts but then describes only two. Step one (I guess this refers to the work undertaken and presented in Section 2) needs to be better elaborated. More info should be provided on how the survey was performed (as well providing the matrix and the questionnaire) – including how many specialists were involved and what is meant with the word specialists. Overall, the method applied needs to be provided in more detail to help the reader understand and increase replicability.

4. Result

For each Criterion in Table 1, there are several sub-indicators applied. How were these extracted? Were these defined by the author? If yes – more details should be provided on how these have been selected – what they represent – If not – then this is mostly a use-case paper (an already available methodology is used) thus all sections should be significantly strengthened to increased added value – knowledge contribution of this work. Some of these sub-indicators are not very clear – a brief explanation should be also provided.

It is not clear how the scores were extracted in Table 1 – is this the average of different responses? How the different responses have been handled? How was the total score extracted (e.g. (10/30 = 33% - this is not clear – please explain). Authors should provide the formulas to extract the total score, city sustainability ration etc. in the Method section.

5. Discussion

Author should attempt to provide some more in-depth justification on why some parameters were scored low or high. E.g., In criterion 3 why are all these low in most criteria?

6. Conclusion

It would be interesting for the author to mention as well whether this process should be applied in annual basis for evaluation (what is the recommended monitoring interval) and whether this method-matrix can be applied as it is by other cities. If this is the case it would be interesting to compare results with other cities.

Author Response

major revisions are made to the article addressing all your comments

Round 2

Reviewer 1 Report

The Author's answers do not contain detailed explanations of the questions asked in the review. The Author sends the Reviewer to re-read the entire paper and to search for answers on their own. The Author's approach to review is inappropriate. That's why I'm asking the same questions again and expecting specific answers.

- lines 79 and 84: It says "Five types ..." and it says ... 6 types, because in line 84 there are two types together, i.e. "Human capital" and "Manufactured capital";

- there is a lack of a broader review of the literature in terms of other similar case studies and the methods that are used during such assessments;

- in section 3. Methods there is no information on what the questions in the questionnaire looked like?, how many respondents answered these questions (what was the size of the research sample)?, what technique was used to conduct the research? how were respondents selected? what is the credibility of the respondents, i.e. how was their knowledge and competence assessed during the research? how was the score (low, medium, high) determined from the responses received in the questionnaires?

- there is no justification that the presented results bring new knowledge to the current state of science.

Author Response

Reply to Reviewer 1

Thank you for your valuable comments, which helped develop this work significantly.

Each of your comments has been addressed thoroughly, as explained below.

The topic of the paper is current. However, significant shortcomings in its content were noticed.

  • lines 79 and 84: It says "Five types ..." and it says ... 6 types, because in line 84 there are two types together, i.e. "Human capital" and "Manufactured capital"; -

This reference is re-written from line 54 as follows:

Considerations of long-term sustainability are important to have a wider view than merely environmental protection.  This approach acknowledges that investing in human, natural, and intellectual capital is frequently neglected in favor of an emphasis on economic or financial value   [6, 7].         

  • there is a lack of a broader review of the literature in terms of other similar case studies and the methods that are used during such assessments; -

Added literature review (from line 78)

Sustainability assessment for cities requires carefully tailored methods to assess urban sustainability performance according to their local or national context. The merit of the assessment methods is evaluated in practice in cities [14, 15]. Several sets of indicators for urban sustainability are identified in the literature, for example, emergy-based indicators of ecosystem services in Simao city in China [54]; a green spaces assessment approach to health, safety, and environment [48]; construction and city related sustainability indicators in Europe [18]; infrastructure sustainability indicators in South African [19]; smart sustainable city indicators [55, 56]. However, no one set of indicators applies to all cities or communities equally [52, 53, 54]. 

  1. Mori K., Christodoulou, A., 2012, Review of sustainability indices and indicators: Towards a new City Sustainability Index (CSI), Environmental Impact Assessment Review, 32(1): 94-106, doi.org/10.1016/j.eiar. 2011. 06.001.

53.Tomah, A., Abed, A., & Saleh, B., 2017, Assessment of the Geographic Distribution of Public Parks in the City of Amman. European Journal of Scientific Research, Vol. 144 No 3 March, 262 - 275.

  1. Pan, Y., Zhang, B, Wu, Y., Tian, Y., 2021, Sustainability assessment of urban ecological-economic systems based on energy analysis: A case study in Simao, China, Ecological Indicators, Vol. 121, 107157, doi.org/10.1016/j.ecolind.2020.107157.
  2. Sumeyye Kusakci, Mustafa K. Yilmaz, Ali Osman Kusakci, Samba Sowe, Fatuma Abdallah Nantembelele, 2022, Towards sustainable cities: A sustainability assessment study for metropolitan cities in Turkey via a hybridized IT2F-AHP and COPRAS approach, Sustainable Cities and Society, Vol. 78, 103655, https://doi.org/10.1016/j.scs.2021.103655

Section 3. Methods

  • there is no information on what the questions in the questionnaire looked like. what technique was used to conduct the research? how many respondents answered these questions (what was the size of the research sample)?

this note is addressed from line 179

Assessment indicators are proposed to quantify the sustainability assessments of Amman city. To this end, candidate indicators are selected after reviewing various materials issued by the local institution, such as Amman Municipality, and international organizations, such as the UN organizations (UN-Habitat, ISO, OECD). The SDG most relevant to urban development that focuses on cities is SDG11: “Make cities and human settlements inclusive, safe, resilient and sustainable, containing nine categories:” [44, 45].

A checklist survey was created according to the nine standard indicators of SDG 11 to assess the sustainability of the city of Amman.  The survey questions are formulated according to each indicator sub-item (appendix 1).  The evaluation of each item consists of three levels: low, average, and optimal.  The checklist was distributed to 10 respondents from different departments of the Municipality of Amman, the Amman Urban Observatory, and the UN-Habitat Jordan office

  • how were respondents selected? what is the credibility of the respondents, i.e. how were their knowledge and competence assessed during the research?

this note is addressed from line 195

The respondents were selected from the Municipality of Amman, the Amman Urban Observatory, and the UN-Habitat Jordan office. The respondents prepared the Voluntary Local Review (VLR) for the City of Amman report in 2022. They were asked to evaluate the sub-indicators of Goal 11 and give points to match the situation in the city and evaluate the performance level in three levels or leave it blank without an answer to reduce error. The indicator values are calculated following data collection, and the initial values are organized into tables and aggregated for each indicator.  The sum of the responses for each sub-indicator is placed in the designated level under L, A, or O (Appendix 1). For example, if there are three sub-indicators, six replies are in the low-level box of the first sub-indicator, four in the low-level box of the second sub-indicator, and two in the third, the average= 6+4+2=12÷ 3= 4. Similar procedure to calculate averages for levels A and O is conducted.  The sum of all values of indicators is presented in Table 1.

  • how was the score (low, medium, high) determined from the responses received in the questionnaires?

The informants mark the appropriate level of performance of indicators in the checklist according to their opinion, in low-performance level, average, and optimal or leave blank

  • there is no justification that the presented results bring new knowledge to the current state of science

the extensive literature review conducted in this study finds no pilot sustainability assessment of the cities in Jordan that shows clear and quantitative assessment data. Even the VLR 2022 report of Amman city and the Green City Action Plan 2018 lack a comprehensive indicator assessment. The methodology applied in this study attempted to fill in this gap.

Reviewer 2 Report

The article has been revised according to the reviewer's suggestions. I make no further comments.

Author Response

-

Reviewer 3 Report

The paper has been improved and some of my comments haven been addressed, however many of them have been poorly or not at all been addressed. Below I highlight the most important ones that need to be reconsidered.

·       Abstract still needs to be strengthened highlighting more the methodology e.g., criteria extracted, info on checklist, score-results.

·       The text still requires some good polishing on the use of English – there are several syntactical and grammatical errors here and there.

·       The literature review has been significantly enhanced, but I still feel that the majority of references is outdated – many tools and frameworks have been published after 2020 and can be found in open access journal (e.g., Smart Cities by MDPI).

·       Method section still needs some clarifications (on how the survey was conducted, providing the questionnaire, what are these specialists). Who is the developer of this method? CASBEE-193 City worldwide-use version, 2011? (Line 193). Where is this reference?

·       My comments from previous round regarding Results, Discussion and Conclusion are only partially addressed.

I would strongly suggest the author to provide more details on how some of these comments have been addressed (not just mentioning generic claims like “the section was revised according to these comments” or “note taken into consideration”), and for those who did not – a quick justification why is that. This will significantly help future evaluations, since now the reviewer needs to identify on his own if and how each comment has been addressed. This is unprofessional. The author should provide detail answers also to my comments from the previous round.

Author Response

Reply to Reviewer 3

Thank you for your valuable comments, which helped develop this work significantly.

Each of your comments has been addressed thoroughly, as explained below.

Abstract:

Your notes: I do not see the added value of the first lines (9-15) appearing on the abstract - since this is general background information that better fits in the Introduction. I suggest discarding this info and instead strengthening the methodology and results-related info in the abstract (e.g., criteria extracted, info on checklist, score-results).

Lines 9-15 are deleted, and the abstract is rewritten highlighting the methodology, extracted criteria, and checklist score and results.  Highlighting changes made to the abstract is not practical as it is completely rewritten.

Accelerated urbanization causes an increasing number of city dwellers, insufficient and overburdened infrastructure and services, and a negative environmental and climate change impact. Measuring the city’s progress toward sustainability is important to support decision-making and policy development.

This study aims to establish an assessment and monitoring method of sustainable development goals at the city level, focusing on identifying indicators compatible with the city context to update and monitor progress toward sustainability.  Existing literature on sustainability assessment methods and tools is reviewed. A comprehensive framework for city sustainability assessment and a checklist of indicators are suggested related to the context of Amman city in Jordan.     A Voluntary Local Review (VLR) report of Amman was presented to the United Nations in 2022. It reviews its progress toward achieving the SDGs. This report, however, lacks clear indicators and a quantitative assessment of the city’s sustainability, which this paper seeks to address.  The checklist survey questions are formulated according to the sub-indicators of the UN-Habitat SDG indicator metadata.  The checklist was distributed to respondents from the Municipality of Amman and related organizations responsible for preparing the VLR.  The respondents evaluated the sub-indicators of Goal 11 and gave performance level scores in three levels: low, average, and optimal. The sum of indicator values is quantitatively presented in tables. The findings reveal indicator values of the city sustainability assessment framework, as applied in this paper, that can be adjusted within the characteristics and constraints of the local context in a two-year observation period to provide updated data for decision-makers regarding the current status and future implementation of sustainability agendas.

Your note: The text requires some good polishing on the use of English – there are several syntactical and grammatical errors here and there e.g., line 16, 23 – the same applies for the rest of the document.

English editing is conducted for all the manuscript, including lines 16 and 23

Your note: The literature review has been significantly enhanced, but I still feel that the majority of references is outdated many tools and frameworks have been published after 2020 and can be found in open access journal (e.g., Smart Cities by MDPI).

Added literature after 2020:

  1. Pan, Y., Zhang, B, Wu, Y., Tian, Y., 2021, Sustainability assessment of urban ecological-economic systems based on energy analysis: A case study in Simao, China, Ecological Indicators, Vol. 121, 107157, doi.org/10.1016/j.ecolind.2020.107157.
  2. Sumeyye Kusakci, Mustafa K. Yilmaz, Ali Osman Kusakci, Samba Sowe, Fatuma Abdallah Nantembelele, 2022, Towards sustainable cities: A sustainability assessment study for metropolitan cities in Turkey via a hybridized IT2F-AHP and COPRAS approach, Sustainable Cities and Society, Vol. 78, 103655, https://doi.org/10.1016/j.scs.2021.103655.
  3. Pira, M. A novel taxonomy of smart sustainable city indicators, 2021, Humanit Soc Sci Commun 8, 197. https://doi.org/10.1057/s41599-021-00879-7
  4. The Ministry of Environment, Water Quality Report, 2021.

http://moenv.gov.jo/EN/List/Annual_Reports

  1. The Ministry of Environment, Ambient Air Quality Monitoring Report,

http://moenv.gov.jo/EN/List/Annual_Reports

  1. UN-Habitat, World Cities Report 2022: Envisaging the Future of Cities.

https://unhabitat.org/world-cities-report-2022-envisaging-the-future-of-cities.

  1. UN-HABITAT, SDG Indicator Metadata, 2022, extracted from: https://unhabitat.org/sites/default/files/2022/08/sdg_indicator_metadata-11.a.1.pdf
  2. Brodny, JarosÅ‚aw, and Magdalena Tutak. 2023. "Assessing the Energy and Climate Sustainability of European Union Member States: An MCDM-Based Approach" Smart Cities6, no. 1: 339-367. https://doi.org/10.3390/smartcities6010017.
  3. Choi, Hee-Sun, and Seul-Ki Song. 2023. "Direction for a Transition toward Smart Sustainable Cities based on the Diagnosis of Smart City Plans" Smart Cities6, no. 1: 156-178. https://doi.org/10.3390/smartcities6010009
  4. Conti, Stefania, Álvaro Dias, and Leandro Pereira. 2023. "Perceived City Sustainability and Tourist Behavioural Intentions" Smart Cities6, no. 2: 692-708. https://doi.org/10.3390/smartcities6020033

  1. Introduction:

The introduction needs to be significantly enhanced. Author should clearly define what is meant by the term sustainability and especially sustainable city (some hints are available in Section 2).

From line 44:

A sustainable city is a city designed with consideration for social, economic, environmental impact and resilient habitat for existing populations without compromising the ability of future generations to experience the same [63]. 

I would expect to see a much more detailed reference to similar works available in literature, including already available tools to assess sustainability and quick reference to results (which cities evaluated etc.).

Added literature review (from line 69)

Sustainability assessment for cities requires carefully tailored methods to assess urban sustainability performance according to their local or national context. The merit of the assessment methods is evaluated in practice in cities       [14, 15]. Several sets of indicators for urban sustainability are identified in the literature, for example, emergy-based indicators of ecosystem services in Simao city in China [54]; a green spaces assessment approach to health, safety, and environment [48]; construction and city related sustainability indicators in Europe [18]; infrastructure sustainability indicators in South African [19]; smart, sustainable city indicators [55, 56]. However, no one set of indicators applies to all cities or communities equally [52, 53, 54]. 

  1. Mori K., Christodoulou, A., 2012, Review of sustainability indices and indicators: Towards a new City Sustainability Index (CSI), Environmental Impact Assessment Review, 32(1): 94-106, doi.org/10.1016/j.eiar. 2011. 06.001.

53.Tomah, A., Abed, A., & Saleh, B., 2017, Assessment of the Geographic Distribution of Public Parks in the City of Amman. European Journal of Scientific Research, Vol. 144 No 3 March, 262 - 275.

  1. Pan, Y., Zhang, B, Wu, Y., Tian, Y., 2021, Sustainability assessment of urban ecological-economic systems based on energy analysis: A case study in Simao, China, Ecological Indicators, Vol. 121, 107157, doi.org/10.1016/j.ecolind.2020.107157.
  2. Sumeyye Kusakci, Mustafa K. Yilmaz, Ali Osman Kusakci, Samba Sowe, Fatuma Abdallah Nantembelele, 2022, Towards sustainable cities: A sustainability assessment study for metropolitan cities in Turkey via a hybridized IT2F-AHP and COPRAS approach, Sustainable Cities and Society, Vol. 78, 103655, https://doi.org/10.1016/j.scs.2021.103655

The author is mentioning that ‘Although many cities have proposed different strategies to implement sustainable practice, yet, the degree of these practices need to be deliberate’. This needs to be justified. Why is there a need to develop a method to assess a city’s sustainability? How about current methods – what are their pros and cons?

This note is addressed in the text from line 57 and from line 89

Although different strategies for conducting sustainable practices have been proposed by many cities, measuring  the progress of practices towards sustainability is also important to support decision-making, and policy development and measure the impact on the environment and context [8].

From Line 89

Establishing comparable indicators may enable cities to share and disseminate effective tools and disseminate on a common network, with a focus on the city scale, as this is the level where the application of urban sustainability indicators can be best appreciated and compared [17, 18].  

  1. Indicators of city sustainability

I do not understand the scope of Section 2 (why a separate section is there). It makes sense for example if the author plans to select, extract etc. his/her own KPIs but the flow of information is not very clear – the story telling needs to be enhanced (e.g., why the five types of value are mentioned in line 79? Why the Green City Index is specifically described? – and not any other of the various similar Indexes available?) these sections are deleted are rewritten

Since the evaluation is based on the nine indicators imposed by SDG Goal 11, it is very important to justify and better support the reasoning behind this choice.

This note is addressed from line 179

Assessment indicators are proposed to quantify the sustainability assessment of Amman city. To this end, candidate indicators are determined following a review of the materials issued by the institutions that produced the VLR of Amman city, which are: the Municipality of Amman, UN-Habitat, UNESCWA. SDG11 is most relevant to urban development that focuses on cities.

  1. Method

This section needs significant improvement as well. The author mentions three parts but then describes only two. Step one (I guess this refers to the work undertaken and presented in Section 2) needs to be better elaborated. More info should be provided on how the survey was performed (as well providing the matrix and the questionnaire) – including how many specialists were involved and what is meant with the word specialists. Overall, the method applied needs to be provided in more detail to help the reader understand and increase replicability.

Who is the developer of this method? CASBEE-193 City worldwide-use version, 2011? (Line 193). Where is this reference?

This reference is now missioned in the review as ‘indicators compatible with the sustainable development goals (line150)

This note is addressed in the text Method section from line 165

Amman's progress in achieving SDG11 is presented in Chapter 2 of the Voluntary Local Review (VLR) prepared by the city municipality in 2020, addressing two indicators only: housing and transportation, and two sub-indicators of the environmental impact of the city, waste management and Air Quality. the Amman VLR lacks a clear and accurate quantitative assessment of the indicators [43].  This paper addresses this deficiency and proposes a methodology for assessing all SDG 11 indicators and providing a more comprehensive sustainability assessment.

Method section still needs some clarifications (on how the survey was conducted, providing the questionnaire, and what these specialists are).

The checklist questions are provided in Appendix 1

It is not clear how the scores were extracted in Table 1 – is this the average of different responses? How the different responses have been handled? How was the total score extracted (e.g. (10/30 = 33% - this is not clear – please explain). Authors should provide the formulas to extract the total score, city sustainability ration etc. in the Method section.

This note is addressed in the text below -From Line 176

Assessment indicators are proposed to quantify the sustainability assessment of Amman city. To this end, candidate indicators are determined following a review of the materials issued by the institutions that produced the VLR of Amman city which are: the Municipality of Amman, UN-Habitat, UNESCWA. SDG11 is most relevant to urban development that focused on cities: “Make cities and human settlements inclusive, safe, resilient and sustainable, containing nine categories” [44, 45]. A checklist survey was created according the nine indicators of SDG 11 to comprehensively assess the sustainability of the city Amman.  The survey questions in Appendix 1 are formulated according to each sub-indicator items, which are subtracted from the SDG indicator metadata 2021 and the World Cities Report, 2022 [59, 60].  The evaluation of each item consists of three performance levels: low, average, and optimal.  The checklist was distributed to 10 key informants from the Municipality of Amman, UN-Habitat Jordan office, UNESCWA, and Amman Urban Observatory, who are familiar with the VLR 2022 of Amman. The evaluation of each indicator is dependent on the value of the sub-indicators. The respondents were asked to evaluate the sub-indicators of the nine indicators of Goal 11 and give scores to match the situation in the city and evaluate the level of performance in the three levels or leave blank to questions they don’t respond to, in order to reduce error.  The indicator values are calculated following data collection and the initial values are organized into tables and aggregated for each indicator.  The sum of the responses for each sub-indicator is placed in the designated level under L, A, or O (Appendix 1). For example, if there are three sub-indicators, 6 replies are in the low-level box of the first sub-indicator, 4 in the low level box of the second sub-indicator, and 2 of the third, the average= 6+4+2=12÷ 3= 4. Similar procedure to calculate averages for levels A and O is conducted.  The sum of all values of indicators is presented in Table 1.

  1. Result

For each Criterion in Table 1, there are several sub-indicators applied. How were these extracted? Were these defined by the author? If yes – more details should be provided on how these have been selected – what they represent – If not – then this is mostly a use-case paper (an already available methodology is used). All sections of the result need to be strengthened and made clear to increase added value – knowledge contribution of this work.

The survey questions in Appendix 1 are formulated according to each sub-indicator items, which are subtracted from the SDG indicator metadata 2021 and the World Cities Report 2022.   [Line 186]

All sections of the Result have be strengthened and made clear to increase added value – knowledge contribution of this work.– and a brief explanation is provided. For example, the first indicator:

  1. Accessibility to suitable, secure and inexpensive housing, essential services, and shanty areas improvement

The assessment result of the first indicator access to decent, safe, and inexpensive housing, essential utilities, and improvement of shantytowns is average-low, however, the proportion of population living in slum households is less than country’s proportion of 18% [58].  The assessment of eligible population ratio included in national social security is low-average. The assessment level of encounters with a qualified professional in a medical facility or in the community per capita per year is average-low, perhaps because many. The level of residents using safely managed water services in urban areas is relatively average, as most households are connected to Amman’s main municipal water supply. In contrast, the level of residents using securely managed sanitation facilities is low as some houses are not connected to the main sanitation facility. The ratio of residents using modern cooking solutions is average as the cooking solutions are available in the market at affordable costs for all.  

  1. Discussion

The author should attempt to provide some more in-depth justification on why some parameters were scored low or high. E.g., In criterion 3 why are all these low in most criteria? This is provided in the discussion of all parameters, an example in line 227 :

  1. 3. Impact on environment assessment level is low including the sub-indictors water sustainability, air sustainability and waste management. Although, the assessment of the sub-indicator of water pollution reduction is average because special attention is made to the water quality of Amman. Houses have access to a municipal water supply once a week, and outages can last for several weeks, especially in the long dry summer season. Additional water tanks are needed for houses to maintain water supplies during water cut-offs and to purchase water when it runs out. In addition, water leakage from the main water network is high, while households suffer from water scarcity.

  1. Conclusion

It would be interesting for the author to mention as well whether this process should be applied in annual basis for evaluation (what is the recommended monitoring interval) and whether this method-matrix can be applied as it is by other cities. If this is the case it would be interesting to compare results with other cities.

These comments are addressed in conclusion section from Line 321

It is recommended that the city establish a monitoring period to assess improvements. In order for cities to understand their current sustainable conditions, a comparison can be made to other cities in the state. The assessment method proposed in this study is expected to assist future sustainability assessments to move towards an increasingly sustainable urban future for all in the city of Amman and could be applied in other cities, with some modifications according to the local context.

Although Jordan provides baseline data from its regional and country levels for the global assessments undertaken every two years to inform international UN agencies of the State of National Urban Policy (NUP) and updated yearly since 2018, a comprehensive local city assessment of SDG11 is endeavored in this study.

Monitoring and reporting of the indicator need to be repeated at annual intervals, and comprehensive reporting undertaken once every two years and made available online on urban policy platforms.

Round 3

Reviewer 1 Report

The author's additions and explanations are satisfactory.

Reviewer 3 Report

The author has adequately adressed most of my comments.